# Does Older Age Modify Associations between Endocrine Disrupting Chemicals and Fecundability?

**DOI:** 10.3390/ijerph19138074

**Published:** 2022-06-30

**Authors:** Anna Z. Pollack, Jenna R. Krall, Shanna H. Swan, Germaine M. Buck Louis

**Affiliations:** 1Department of Global and Community Health, College of Health and Human Services, George Mason University, Fairfax, VA 22030, USA; jkrall@gmu.edu (J.R.K.); glouis@gmu.edu (G.M.B.L.); 2Department of Preventive Medicine and Public Health, Icahn School of Medicine at Mount Sinai, New York, NY 10029, USA; shannahswan@gmail.com

**Keywords:** bisphenol, benzophenone, couples, endocrine disrupting, effect modification, fertility, phthalate, ultraviolet filter

## Abstract

Urinary concentrations of several endocrine disrupting chemicals, including phthalate metabolites, bisphenol A (BPA), and benzophenone (BP)-type ultraviolet (UV) filters, have been associated with a longer time-to-pregnancy (TTP). Potential modification of these associations by couple’s age has not been studied. TTP was defined as the number of prospectively observed menstrual cycles a couple attempted pregnancy until the occurrence of a human chorionic gonadotropic-detected pregnancy. Urinary concentrations of two BP-type UV filters and three phthalate metabolites were measured at baseline. Fecundability odds ratios (FORs) and 95% confidence intervals (CIs) were estimated for each chemical adjusting for age, body mass index, serum cotinine, creatinine, and accounting for right censoring and left truncation. Models evaluated effect modification between EDC concentrations and TTP by partner’s age, dichotomized at 35 years. Separate models were run for male and female partners. No significant effect modification was observed for any EDC for either partner, but data were suggestive of a longer TTP among females aged ≥35 years, particularly for BP-2 (FOR = 0.61, 95% CI 0.36, 1.05) and 4-hydroxybenzophenone (FOR = 0.71, 95% CI: 0.46, 1.09) reflecting 39% and 29% reductions in fecundability, respectively. We saw no evidence of effect modification by couples’ age on associations between TTP and urinary phthalate or BPA metabolite concentrations. Across the EDCs we examined, we found little evidence that age modifies TTP-exposure associations.

## 1. Introduction

Human exposure to endocrine disrupting chemicals (EDCs) is widespread and may contribute to reproductive health concerns [1]. For example, phthalates are plasticizers used in a variety of consumer products including personal care products, polyvinyl flooring, feminine hygiene products, and food packaging [2,3,4,5]. Bisphenol A (BPA) is a plasticizer that is found in cash register receipts, dental sealants, and certain plastics. Benzophenone (BP)-type ultraviolet (UV) light filters are constituents in cosmetics, plastics, and printing inks, among other commercial uses [6]. Dermal application is the primary route of exposure for benzophenones, with high skin penetration [1,2,3,7,8,9,10]. Phthalate exposure commonly occurs through dermal exposure and via food consumption [4,11]. Human exposure to phthalates is ascertained by measurement of monoesters in urine, as is exposure to benzophenones [12,13].

Despite short half-lives for phthalates and the environmental phenols BPA and benzophenone UV filters, populations are exposed through repeated use of personal care products and other commercial products, and may experience continual exposure [5,12,14]. Mechanistic research indicates both estrogenic and anti-androgenic properties for benzophenone ultra violet (UV) light-type filters [15] and also an ability to bind with the estrogen receptors [16]. Phthalates have both estrogenic and anti-androgenic properties and have been associated with reduced fecundity and changes in reproductive hormones across the menstrual cycle [17,18,19,20,21,22]. Similarly, BPA is estrogenic [23]. A number of modifiers of EDC-health associations have been examined (including stress and race/ethnicity) but we are unaware of research that has assessed potential modifying effects of couples’ ages. In particular we examined this potential modification relative to EDC associations with fecundability, defined as the per cycle probability of pregnancy.

EDCs have been associated with a range of fecundity endpoints in women and men, including TTP [24,25,26]. Recent reviews note that there is inconsistent and scant evidence for phthalates, BPA, and ultraviolet filters in relation to time to pregnancy [26,27,28], especially based upon prospective cohort studies designed to capture observed time trying or at risk for pregnancy such as the LIFE Study. Longer TTP may serve as an indicator of worse overall health status [29]. Therefore, fecundability reflects a sensitive reproductive endpoint with important later-in-life health implications. Evidence from prospective pregnancy cohort studies that included couples recruited from the general population indicate that diisononyl phthalate [18], monoethyl phthalate [30], and triclosan [31] in female partners were associated with longer TTP. Among male partners, benzophenone-2 and 4-hydroxybenzophenone were associated with a longer TTP [32]. Among infertile couples undergoing assisted reproductive technologies, phthalates were associated with lower oocyte yield, failed clinical pregnancy, and decreased implantation success [19,33].

Age is a known determinant of fecundability [34] and EDC metabolism is affected by age [35]. The potential modification of associations between EDCs and reproductive endpoints by age is plausible. While most research to date has adjusted for age, none has examined the potential modifying effect of age. Therefore, the purpose of this study was to determine if age modifies the association between EDCs and fecundability as measured by TTP. Our hypothesis is that at older ages, EDCs will be more strongly associated with a longer TTP relative to younger ages.

## 2. Materials and Methods

### 2.1. Study Population

The Longitudinal Investigation of Fertility and Environment (LIFE) Study comprised a prospective cohort of 501 couples discontinuing contraception to try for pregnancy, and who resided in 16 counties in Michigan and Texas between 2005–2009. Population-based recruitment strategies were implemented as described previously [36]. The inclusion criteria were: couple in a committed relationship; female partner aged 18–44 years and male partner aged 18 years and older; fluent in English or Spanish; and the absence of physician-diagnosed infertility history in either partner. Additional inclusion criteria for women were menstrual cycles between 21–42 days, no use of injectable hormonal contraceptive within the prior year, and not currently lactating. Human subjects research approval was obtained from all institutions and all study participants provided full informed consent prior to data collection.

### 2.2. Data Collection

Couples were screened for eligibility and enrolled in the cohort. Baseline interviews captured covariates such as age (years) followed by measurement of height and weight to estimate body mass index (BMI, kg/m^2^), the collection of blood and urine collection for quantification of serum cotinine (ng/g serum), and non-persistent EDCs (mg/dL), respectively. Female participants were provided with Clearblue^®^ Easy (Swiss Precision Diagnostics, Geneva, Switzerland) fertility monitors and instructed on their use per the manufacturer’s guidelines, which indicate starting monitoring for ovulation on day 6 of the cycle. The fertility monitors track daily levels of estrone-3-glucuronide and luteinizing hormone and their ratio corresponds with monitor indications of low, high, and peak fertility. Home pregnancy tests (Clearblue^®^ Easy) were provided to women at enrollment. Women were instructed to test each cycle on the day that they expected menses. Cotinine concentration (ng/mL) was measured in 1 mL of serum by liquid chromatography-isotope dilution tandem mass spectrometry. Additionally, urinary creatinine (mg/dL) was measured to account for urine dilution when quantifying EDCs. Couples completed daily journals that captured sexual intercourse and menses (for females), which augmented monitor data to ascertain TTP.

### 2.3. Chemical Measurement

The Wadsworth Center (Albany, NY, USA) quantified non-persistent chemicals that were previously studied in relation to TTP in this cohort [17,32]. Urinary concentrations of bisphenol A (BPA) (ng/mL) were measured by high-performance liquid chromatography (HPLC) coupled with API 2000 electrospray triple-quadrupole mass spectrometer (MS/MS) [37]. The laboratory limit of quantitation (LOQ) was 0.05 ng/mL, which was twice the lowest valid acceptable calibration standard. Phthalate metabolites were measured in urine by HPLC-MS/MS as described previously [13]. The following phthalate metabolites were measured: monobenzyl phthalate (mBzP), mono-n-butyl phthalate (mBP), mono(2-ethyl-5-carboxylpentyl) phthalate (mCEPP), mono-[(2-carboxymethyl)hexyl] phthalate (mCMHP), mono (3-carboxypropyl) phthalate (mCPP), monoethyl phthalate (mEP), mono(2-ethyl-5-hydroxyhexyl) phthalate (mEHHP), mono(2-ethyl-5-oxohexyl) phthalate (mEOHP), mono(2-isobutyl phthalate (miBP), monomethyl phthalate (mMP), monocyclohexyl phthalate (mCHP), mono (2-ethylhexyl) phthalate (mEHP), mono-isononyl phthalate (mNP), and monooctyl phthalate (mOP). Benzophenone (BP)-type ultraviolet (UV) light filter chemicals included BP-3 and its metabolites, 2,4-dihydroxybenzophenone (2,4-OH-BP) (also called BP-1), 2,2′-dihydroxy-4-methoxybenzophenone (2,2′-OH-4-MeO-BP or BP-8), 2,2′,4,4′-tetrahydroxybenzophenone (also called BP-2), and 4-hydroxybenzophenone (4OH-BP). BP-type UV filters were measured with high-performance liquid chromatography–triple-quadrupole tandem mass spectrometry using established methods [7]. Creatinine (mg/dL) was measured with a Roche Hitachi 912 Chemistry Analyzer (Roche Diagnostics Corporation, Dallas, TX, USA) with the Creatinine Plus Assay (Roche Diagnostics Corporation, Indianapolis, IN, USA).

### 2.4. Statistical Methods

Descriptive statistics were conducted as follows. We first examined the distributions for all variables. Continuous covariates (age, BMI, cotinine, and creatinine) and right-skewed chemicals were summarized using their median and interquartile range (IQR). We calculated the percentage of EDC concentrations below the LOD to determine those meeting the requirement of <50% below LOD for analysis. The phthalate metabolites mOP, mNP, mCHP, mMP, and mEHP had >50% values below the LOD and were not included in the main analysis.

We fitted discrete time Cox proportional hazards models [38], which allow for a cycle-varying intercept. These models accounted for left truncation given the uncertainty in the time couples may have been at risk for pregnancy before enrollment in the study. Couples were censored at study withdrawal or after 12 months of trying to become pregnant. Models were fitted separately for male and female partners to identify whether age modified associations between partner-specific EDC concentrations and TTP. All EDC concentrations were log transformed and standardized by their standard deviations to aid in the interpretation of point and interval estimates, given their small unit of analysis [17].

All models included an interaction term between each age category and study chemicals and effect modification was evaluated by comparing FOR of chemicals between age strata. Age was categorized as <35 years or ≥35 years to associations in women meeting the clinically relevant designation of advanced maternal age (≥35 years) [39]. Associations are reported as fecundability odds ratios (FORs) with associated 95% confidence intervals (CI). FORs < 1 indicate a reduced probability of cycle-specific pregnancy or diminished fecundability reflecting in a longer TTP. FORs > 1 indicate enhanced fecundability reflecting a shorter TTP. To improve interpretability of effect modification results, FORs were reported for each chemical by age category (<35 and ≥35 years). Confounders were selected based on a directed acyclic graph informed by prior relevant studies. Previous work reported that cigarette smoking was associated with a pronounced (~50%) reduction in fecundability, resulting in a longer TTP [40]. Cotinine levels are associated with fecundability and are also associated with levels of EDCs. Models were adjusted for age (years, continuous), urinary creatinine (mg/dL), serum cotinine (active smoking > 10 ng/mL vs. non-active smoking ≤ 10 ng/mL), and BMI (weight in kg/height in m^2^).

### 2.5. Sensitivity Analyses

To determine whether our results were sensitive to the specific form of our age effect modification term, we compared our models to those with ages in sex-specific quartiles to explore potential nonlinearity. We also fit models controlling for both partners’ chemical concentrations (i.e., models for female partners included adjusting for male partner’s concentrations), as well as models with additional adjustment for race/ethnicity and income.

## 3. Results

Our study cohort includes 403 (80%) female and 386 (77%) male partners with complete data and available urine for EDC quantification from the original LIFE Study cohort comprising 501 couples. Missing data for covariates was minimal, ranging from 1% for BMI to 9% for creatinine. Seventy percent of couples had a detectable pregnancy. Among female partners, *n* = 62 (15.4%) were ≥35 years compared to *n* = 115 (29.8%) of male partners. The median ages at study entry were 29 years (IQR = 6) for females and 32 years (IQR = 6) for their male partners and 2.5th and 97.5th percentiles for age was (23, 38) and (24, 42) for females and males, respectively (Table 1). Median BMI was 25.8 kg/m^2^ (IQR = 8.6) for females and 28.8 kg/m^2^ (IQR = 5.9) for males.

Among women ≥35 years (referred to here as older women) the proportion becoming pregnant during the study period varied by urinary concentration of several phthalate metabolites. Higher concentration of mBP, mBzP, and BP-3 and lower levels of mEHHP were seen in older women who did not become pregnant (Table 2). Women < 35 years who did not become pregnant had higher median levels of mEP and BP-3 compared to similarly aged women who became pregnant (Table 2). Men ≥ 35 years of age, whose partner did not become pregnant, had higher median levels of mCMHP and mCEPP compared to similarly aged men whose partners became pregnant (Table 2).

Tests for effect modification showed no significant differences in EDC-TTP associations by age for either partner (Figure 1, Table 3). FORs for phthalate metabolites and BPA differed little by age for either men or women. For BP-type UV filters, little differences in FORs were seen by age for men. However, among women ≥35 years, longer TTP was observed per one unit increase of BP-2 (FOR = 0.61, 95% CI: 0.36, 1.05), 4OH-BP (FOR = 0.71, 95% CI: 0.46, 1.09), BP-8 (FOR 0.80, 95% CI: 0.54, 1.20), BP-3 (FOR = 0.86, 95% CI: 0.60, 1.23), and BP-1 (FOR = 0.91, 95% CI: 0.65, 1.29).

Our results were largely consistent in sensitivity analyses that modeled age in partner specific quartiles, adjusted for the other partner’s chemical concentration, race/ethnicity, and income in separate models. Associations were not substantially changed when we modeled age in partner specific quartiles, although there was an indication of diminished fecundability for the oldest and youngest aged quartiles for females for BP-2, 4OH-BP, and an indication of reduced fecundability for the youngest age quartile for mCEPP, mCMHP, mEP, mEHHP, mEOHP and mCPP (Appendix A). There was no indication of effect measure modification for males when age was modeled in quartiles. The results did not change substantially when we adjusted for the other partner’s chemical concentration (Appendix A) or race/ethnicity and income (results not shown).

## 4. Discussion

In this prospective cohort study of couples followed to incident pregnancy or up to one year of trying, there was not strong evidence of effect modification by age between TTP and urinary concentrations of phthalate metabolites, BPA, or BP-type filters. Among women ≥35 years of age, 4OH-BP and BP-2 were associated with a nearly 30% and 40% reduction in fecundability, respectively, albeit with imprecise confidence intervals. Other BP-type filters including BP-3, BP-1 and BP-8 were also associated with diminished fecundability among women aged ≥35 years, although estimates were imprecise.

We did not identify any other studies that examined age modification of associations between these short-lived EDCs (phthalate metabolites, BPA, and BP-type filters) and fecundability as measured by TTP. We observed that BP-type filters including BP-2, 4OH-BP, BP-1, BP-3, and BP-8 were associated with longer TTP for females ≥35 years, but not for younger women or for men. While BP-type filters have not been fully assessed mechanistically, there is some evidence that they have endocrine disrupting activity. BP-type filters, in particular BP-2, exerted estrogenic effects on both fish sperm and ovary in toxicologic [41]. Biological aging may affect endometrial receptivity and, therefore, could modify the effect of BP-type filters on fecundability. Epigenetic changes occurred following BP-type filter exposure [42,43] and they are reported to increase with age [44]. Therefore, this suggests a possible mechanism by which benzophenones may affect fecundability. In regard to the of the lack of effect modification by age, studies suggest that nutrients such as folate may ameliorate the impact of chemical exposures on fertility outcomes [21,45], which may help explain our findings. In addition, nanoformulation of antioxidants show promise in potentially mitigating the effects of EDCs [46].

Our findings are strengthened by the prospective cohort study that was recruited through population-based sampling methods. This is important, as studies of environmental chemicals in relation to reproductive outcomes focused primarily on biomarkers collected during pregnancy or in populations seeking fertility treatment. Our cohort was intended to include couples with environmentally-relevant EDCs concentrations amongst reproductive aged couples at risk for pregnancy. Urine specimens were collected before couples began trying for pregnancy and use of the fertility monitor increased the validity of TTP as did the use of digital pregnancy kits to capture hCG-pregnancy. Moreover, we observed no differences in the pregnancy status of couples with/without remaining urine samples for EDC quantification. BMI was measured by research assistants and not dependent upon self-reported height and weight, and cigarette smoking status was measured by serum cotinine concentration. The laboratory methods for measuring phthalate metabolites, BPA, and BP-type filters was another strength.

The age range of women in the LIFE Study may be a study limitation and too narrow to capture effect modification, in that women <18 or >40 years were ineligible. While age did not modify the relationship between urinary phthalate metabolite levels and fecundability, the median age of female study participants was 29 years (IQR = 6 years). Only 16% of women in the LIFE Study were aged 35 years or older, which may have decreased precision. Among women ≥35 years of age, the median age was 36, underscoring this limitation. Twice as many men were aged ≥35 (29.8%), and 5% exceeded 41.7 years. Therefore, this study had greater power to address the hypothesis of effect modification by age in men than women. As such, we are more confident in the absence of such effect modification in men than women. Yet, the findings for BP-type filters, BP-1, BP-2, BP-3, BP-8, and 4OH-BP suggested longer TTP among women aged ≥35 years. Although inference among older women in our study is a challenge given the age distribution, this age distribution reflects the proportion of women trying for pregnancy within the age range of 18–40 years. Another important limitation is reliance on a single spot urine in which EDCs were quantified. These are short-lived EDCs that are rapidly metabolized [47] and may not reflect actual exposure during sensitive windows of conception and implantation (both of which can affect TTP). As such, our working assumption is that there is continual exposure in light of the ubiquitous environmental sources for reproductive aged couples. Our study considered effect modification on the multiplicative scale. Future work may explore the possibility of additive interactions [48,49,50].

Synthesizing our findings in the context of other research is challenging due to no other studies examining modification by age in relation to the effects of short-lived chemicals on fecundability in couple-based studies. A recent review identified no additional epidemiologic studies of BP-type filters and fecundability [25], highlighting the need for additional research on these emerging chemicals. Our findings regarding phthalates and BPA are supported by other recent reviews indicating modest or no association with fecundability. Our findings add to the literature in suggesting the absence of strong effect modification by age for these EDCs and human fecundability as measured by TTP. This question, which remains of interest, should be addressed in cohorts adequately powered to examine these risks in women and couples age 40 and older, a demographic that has been increasing.

## 5. Conclusions

This is the first study to examine whether male and female age modifies the association between specific EDCs and fecundability as measured by TTP. Overall, age did not modify the relation between phthalate metabolites, BPA, and BP-type filters and fecundability. For women ≥35 years, there was some evidence that BP-type filters were associated with a longer TTP. Given the limited evidence of effect modification by age on EDC exposure and fecundability, corroboration in sufficiently powered cohorts is needed for a more complete understanding of the modifying effect of age.

## Figures and Tables

**Figure 1 ijerph-19-08074-f001:**
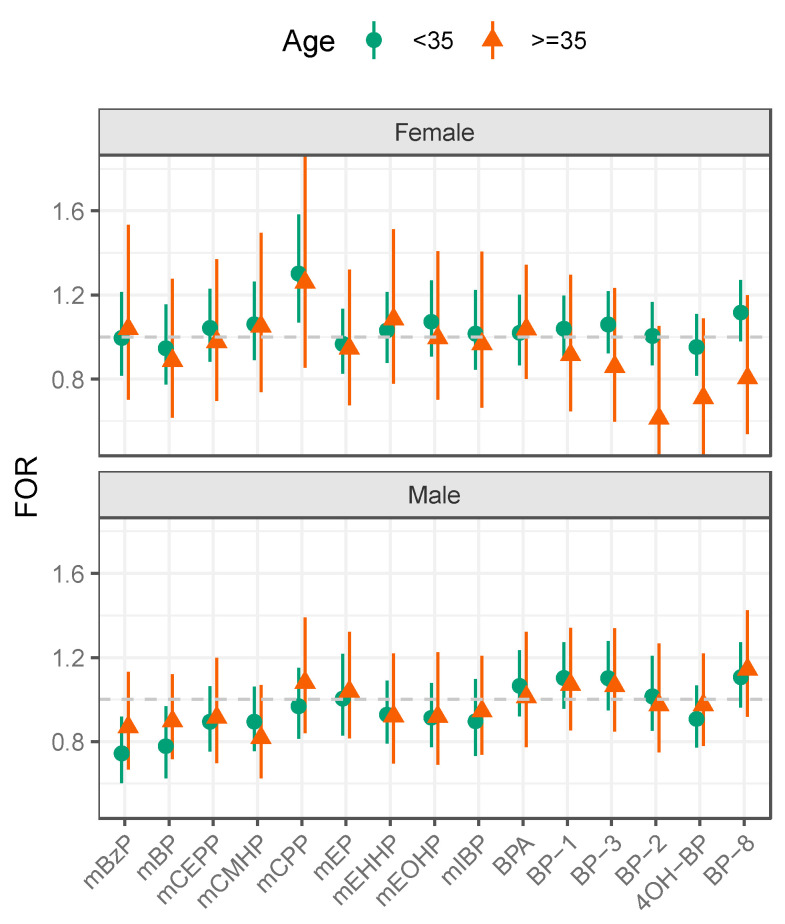
Fecundability odds ratios (FORs) for urinary phthalate metabolites, bisphenol A and benzophenone-type ultraviolet filters by age (<35 to age ≥35 years) among participants in the Longitudinal Investigation on Fertility and Environment Study in Michigan and Texas (2005–2009). NOTE: monobenzyl phthalate (mBZP), mono-n-butyl phthalate (mBP), mono(2-ethyl-5-carboxylpentyl) phthalate (mCEPP), mono-[(2-carboxymethyl)hexyl] phthalate (mCMHP), mono(3-carboxypropyl) phthalate (mCPP), monoethyl phthalate(mEP), mono(2-ethyl-5-hydroxyhexyl) phthalate (mEHHP), mono(2-ethyl-5-oxohexyl) phthalate (mEOHP), mono(2-isobutyl phthalate (miBP), bisphenol A (BPA), 4-hydroxybenzophenone (4OH-BP), benzophenone 1 (BP-1), benzophenone 2 (BP-2), benzophenone 3 (BP-3), benzophenone 8 (BP-8). Models were adjusted for age (years), urinary creatinine (mg/dL), serum cotinine (>10 ng/mL vs. ≤10 ng/mL), and BMI (kg/m^2^).

**Table 1 ijerph-19-08074-t001:** Covariate summary statistics [median (interquartile range)] by pregnancy status, age category, and sex among participants in the Longitudinal Investigation on Fertility and Environment Study in Michigan and Texas 2005–2009.

Variable	Total			<35 Years		≥35 Years	
	Total	Not Pregnant	Pregnant	Not Pregnant	Pregnant	Not Pregnant	Pregnant
**Female partners (*n*)**	**403**	**122**	**281**	**94**	**247**	**28**	**34**
Age (years)	29 (6)	30 (7)	29 (6)	28 (5)	29 (4)	37 (2)	36 (2)
BMI (kg/m^2^)	25.8 (8.6)	27.3 (10.4)	25.2 (7.7)	27.4 (9)	24.8 (7.4)	26.6 (11.9)	27.3 (7.6)
Cotinine (ng/mL)	0.02 (0.08)	0.05 (2.24)	0.02 (0.03)	0.05 (1.68)	0.02 (0.03)	0.09 (13.22)	0.02 (0.04)
**Male partners (*n*)**	**386**	**114**	**272**	**72**	**199**	**42**	**73**
Age (years)	32 (6)	32 (7)	31 (6)	29 (5)	30 (4)	38 (5)	37 (3)
BMI (kg/m^2^)	28.8 (5.9)	29.3 (5.9)	28.6 (5.8)	29.8 (7.2)	28.7 (5.8)	28 (4.3)	28.4 (6.4)
Cotinine (ng/mL)	0.04 (1.36)	0.09 (34.02)	0.03 (0.23)	0.1 (3.26)	0.04 (0.18)	0.08 (153.94)	0.03 (0.71)

**Table 2 ijerph-19-08074-t002:** Phthalate metabolite and ultraviolet filter concentrations [median (interquartile range)] (ng/mL) by pregnancy status, age category, and sex among participants in the Longitudinal Investigation on Fertility and Environment Study in Michigan and Texas (2005–2009).

		Total			<35 Years		≥35 Years	
Chemical (ng/mL)	%<LOQ	Total	Not Pregnant	Pregnant	Not Pregnant	Pregnant	Not Pregnant	Pregnant
**Female partners**		***n* = 403**	***n* = 122**	***n* = 281**	***n* = 94**	***n* = 247**	***n* = 28**	***n* = 34**
mBzP	4	3.7 (7.8)	4.4 (8.3)	3.4 (7.5)	4.4 (7.9)	3.5 (7.4)	4.6 (8.3)	2.9 (7.7)
mBP	1	8.0 (14.5)	9.2 (18.5)	7.8 (14.1)	8.8 (17.4)	7.8 (14.1)	11.6 (24.6)	8.3 (12.8)
mCEPP	2	14.7 (29.5)	15.1 (38.1)	14.7 (27.6)	15.1 (33.8)	14.7 (27.8)	14.2 (43.5)	13.2 (24.6)
mCMHP	1	11.4 (25.5)	11.9 (26.6)	11.2 (22.8)	11.5 (28.2)	11.8 (21.9)	13.1 (20.6)	9.7 (26.8)
mCPP	6	3.9 (8.7)	3.6 (9.8)	4.2 (7.5)	3.6 (9.7)	4.2 (7.4)	3.7 (9.5)	3.7 (8.1)
mEP	2	68.7 (172.0)	89.0 (213.9)	61.7 (156.5)	92.9 (187.9)	62.2 (163.7)	61.9 (264.7)	60.0 (111.7)
mEHHP	2	10.7 (20.9)	10.7 (21.2)	10.5 (20.5)	11.3 (20.9)	10.7 (20.3)	7.0 (20.7)	10.3 (24.8)
mEOHP	4	6.0 (13.2)	5.7 (12.3)	6.0 (13.5)	5.7 (11.0)	6.1 (13.9)	5.5 (19.9)	5.4 (10.9)
mIBP	4	4.0 (7.3)	4.2 (9.1)	3.9 (6.9)	4.0 (8.9)	4.0 (6.9)	4.6 (10.4)	2.1 (6.1)
BPA	2	0.4 (0.8)	0.5 (0.9)	0.4 (0.7)	0.5 (0.9)	0.4 (0.6)	0.4 (0.9)	0.6 (1.5)
BP-1	1	2.5 (12.9)	1.6 (8.4)	2.8 (14.9)	1.2 (6.9)	2.8 (13.7)	5.4 (27.7)	3.3 (19.0)
BP-3	1	5.1 (29.0)	3.6 (27.0)	6.7 (31.8)	2.4 (15.2)	6.8 (30.3)	13.3 (66.7)	5.4 (43.9)
BP-2	28	0.05 (0.15)	0.06 (0.18)	0.05 (0.13)	0.05 (0.15)	0.05 (0.12)	0.11 (0.2)	0.02 (0.16)
4OH-BP	6	0.14 (0.25)	0.14 (0.28)	0.13 (0.24)	0.14 (0.28)	0.13 (0.24)	0.16 (0.58)	0.12 (0.22)
BP-8	29	0.11 (0.65)	0.10 (0.3)	0.12 (0.94)	0.08 (0.26)	0.12 (0.94)	0.16 (0.79)	0.07 (0.87)
Creatinine	–	79.8 (103.0)	88.4 (112.0)	77.2 (99.9)	90.1 (109.6)	80.2 (99.7)	82.3 (118.5)	49.2 (72.0)
**Male partners**		***n* = 386**	***n* = 114**	***n* = 272**	***n* = 72**	***n* = 199**	***n* = 42**	***n* = 73**
mBzP	4	3.7 (7.1)	4.1 (7.9)	3.4 (6.9)	4.8 (8.1)	3.2 (7.0)	3.3 (7.9)	4.1 (6.7)
mBP	1	7.5 (12.0)	8.0 (17.3)	7.1 (10.7)	8.3 (15.7)	6.8 (10.0)	7.4 (18.6)	8.9 (11.6)
mCEPP	1	20.5 (37.7)	22.5 (38.8)	19.6 (37.5)	22.5 (34.7)	21.4 (37.6)	22.4 (48.6)	15.0 (36.0)
mCMHP	0	18.6 (40.5)	21.1 (40.5)	17.2 (40.0)	18.9 (38.6)	19.5 (40.5)	26.5 (41.2)	15.9 (27.9)
mCPP	3	5.6 (9.6)	5.6 (9.0)	5.7 (10.4)	6.2 (8.3)	5.6 (10.4)	3.8 (9.0)	5.7 (9.6)
mEHHP	1	15.4 (33.0)	16.9 (37.2)	15 (29.9)	16.9 (32.2)	16.5 (30.5)	17.4 (38)	11.5 (24.6)
mEP	1	97.2 (261.7)	86.9 (239.1)	102.1 (262.4)	95.1 (256.7)	98.6 (263.5)	50.4 (232.5)	111.7 (250.9)
mEOHP	2	7 (15.2)	7.7 (16.7)	6.8 (14.5)	7.7 (14.8)	6.9 (16.2)	8.2 (19.1)	6.1 (10.6)
mIBP	2	4.5 (7.3)	4.8 (7.6)	4.4 (7.1)	4.5 (7.1)	4.3 (7.2)	5.1 (8.0)	5.4 (6.5)
BPA	2	0.5 (0.9)	0.4 (0.8)	0.6 (0.9)	0.4 (0.7)	0.6 (0.9)	0.4 (0.9)	0.6 (1.3)
BP-1	1	1.3 (8.0)	0.6 (2.3)	1.9 (9.9)	0.6 (2.0)	2.2 (10.0)	0.7 (3.6)	1.3 (9.5)
BP-3	2	3.0 (16.2)	1.7 (5.2)	4.0 (21.5)	1.9 (4.4)	4.1 (24.6)	1.4 (5.8)	3.9 (16.0)
BP-2	28	0.05 (0.12)	0.05 (0.15)	0.05 (0.11)	0.05 (0.11)	0.05 (0.11)	0.03 (0.28)	0.03 (0.13)
4OH-BP	4	0.14 (0.25)	0.13 (0.28)	0.14 (0.24)	0.11 (0.26)	0.15 (0.25)	0.14 (0.28)	0.13 (0.24)
BP-8	27	0.08 (0.44)	0.05 (0.23)	0.09 (0.75)	0.04 (0.25)	0.11 (0.58)	0.07 (0.17)	0.07 (0.91)
Creatinine	–	139.8 (129.9)	115.2 (139.5)	145.4 (126.7)	114.4 (143.3)	146.4 (120.5)	133.9 (96.4)	140 (123.9)

NOTE: monobenzyl phthalate (mBZP), mono-n-butyl phthalate (mBP), mono(2-ethyl-5-carboxylpentyl) phthalate (mCEPP), mono-[(2-carboxymethyl)hexyl] phthalate (mCMHP), mono(3-carboxypropyl) phthalate (mCPP), monoethyl phthalate(mEP), mono(2-ethyl-5-hydroxyhexyl) phthalate (mEHHP), mono(2-ethyl-5-oxohexyl) phthalate (mEOHP), mono(2-isobutyl phthalate (miBP), bisphenol A (BPA), 4-hydroxybenzophenone (4OH-BP), benzophenone 1 (BP-1), benzophenone 2 (BP-2), benzophenone 3 (BP-3), benzophenone 8 (BP-8).

**Table 3 ijerph-19-08074-t003:** Fecundability odds ratios (FORs) for urinary phthalate metabolites, bisphenol A, and benzophenone-type ultraviolet filters by age (<35 to age ≥35 years) among participants in the Longitudinal Investigation on Fertility and Environment Study in Michigan and Texas (2005–2009).

	Females		Males	
Chemical (ng/mL)	<35 Years	≥35 Years	<35 Years	≥35 Years
mBzP	1.00 (0.82, 1.21)	1.04 (0.70, 1.53)	0.74 (0.60, 0.92)	0.87 (0.67, 1.13)
mBP	0.95 (0.78, 1.16)	0.89 (0.62, 1.28)	0.78 (0.63, 0.97)	0.90 (0.72, 1.12)
mCEPP	1.04 (0.88, 1.23)	0.98 (0.70, 1.37)	0.89 (0.75, 1.06)	0.91 (0.70, 1.20)
mCMHP	1.06 (0.89, 1.26)	1.05 (0.74, 1.50)	0.89 (0.75, 1.06)	0.82 (0.62, 1.07)
mCPP	1.30 (1.07, 1.58)	1.26 (0.85, 1.86)	0.97 (0.81, 1.15)	1.08 (0.84, 1.39)
mEP	0.97 (0.83, 1.13)	0.95 (0.68, 1.32)	1.00 (0.83, 1.22)	1.04 (0.82, 1.32)
mEHHP	1.03 (0.88, 1.21)	1.09 (0.78, 1.51)	0.93 (0.79, 1.09)	0.92 (0.70, 1.22)
mEOHP	1.07 (0.91, 1.27)	0.99 (0.70, 1.41)	0.91 (0.77, 1.08)	0.92 (0.69, 1.23)
mIBP	1.02 (0.85, 1.22)	0.97 (0.66, 1.41)	0.90 (0.73, 1.10)	0.94 (0.74, 1.21)
BPA	1.02 (0.87, 1.20)	1.03 (0.80, 1.34)	1.07 (0.92, 1.23)	1.01 (0.78, 1.32)
BP-1	1.04 (0.90, 1.20)	0.92 (0.65, 1.30)	1.10 (0.96, 1.27)	1.07 (0.85, 1.34)
BP-3	1.06 (0.92, 1.22)	0.86 (0.60, 1.23)	1.10 (0.95, 1.28)	1.07 (0.85, 1.34)
BP-2	1.01 (0.87, 1.17)	0.61 (0.36, 1.05)	1.02 (0.85, 1.21)	0.97 (0.75, 1.27)
4OH-BP	0.95 (0.82, 1.11)	0.71 (0.46, 1.09)	0.91 (0.77, 1.07)	0.97 (0.78, 1.22)
BP-8	1.12 (0.98, 1.27)	0.80 (0.50, 1.20)	1.11 (0.96, 1.27)	1.14 (0.92, 1.42)

NOTE: monobenzyl phthalate (mBZP), mono-n-butyl phthalate (mBP), mono(2-ethyl-5-carboxylpentyl) phthalate (mCEPP), mono-[(2-carboxymethyl)hexyl] phthalate (mCMHP), mono(3-carboxypropyl) phthalate (mCPP), monoethyl phthalate(mEP), mono(2-ethyl-5-hydroxyhexyl) phthalate (mEHHP), mono(2-ethyl-5-oxohexyl) phthalate (mEOHP), mono(2-isobutyl phthalate (miBP), bisphenol A (BPA), 4-hydroxybenzophenone (4OH-BP), benzophenone 1 (BP-1), benzophenone 2 (BP-2), benzophenone 3 (BP-3), benzophenone 8 (BP-8). Models were adjusted for age (years), urinary creatinine (mg/dL), serum cotinine (>10 ng/mL vs. ≤10 ng/mL), and BMI (kg/m^2^).

## Data Availability

The data underlying this article will be shared on reasonable request to the corresponding author.

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
