# Peer review of "Does Older Age Modify Associations between Endocrine Disrupting Chemicals and Fecundability?"

_ijerph, 2022, doi:10.3390/ijerph19138074_

Round 1
Reviewer 2 Report
The purpose and hypothesis need to be stated in the introduction.
Why limit the recruitment age of the females but not the males. The fertility of males decreases with older age, which directly influences pregnancy outcomes.
Table 1 and figure 1 legend needs abbreviations.
Reviewer 3 Report
The manuscript titled "Does older age modify associations between endocrine disrupting chemicals and fecundability?" is a very interesting paper in the field of endocrine disrupters. The overall structure is of good quality, introduction and methods are clear and results are of good quality. References are of good quality and cover the principal areas of the research. However, authors should improve the mansucriot in several parts:
1) in the introduction, authors should improve the description of possible increased risk od cardiovascular events in patients vulnerable like cancer patients ( you can cite doi: 10.1016/j.etap.2019.03.006. )
2) authors should explain how some nutraceuticals like curcumin should reduce the side effects of BPA through PPAR and pAMPK pathways.
3) authors should explain how nanoformulation of antioxidants should reduces the side effects of BPA and other endocrine disrupters in preclinical and clinical models ( i.e also in cancer bearing mice). cite 10.1016/j.nano.2016.08.022
